# PRAME Immunohistochemistry in Thin Melanomas Compared to Melanocytic Nevi

**DOI:** 10.3390/diagnostics14182015

**Published:** 2024-09-12

**Authors:** Iulia Zboraș, Loredana Ungureanu, Simona Șenilă, Bobe Petrushev, Paula Zamfir, Doinița Crișan, Flaviu Andrei Zaharie, Ștefan Cristian Vesa, Rodica Cosgarea

**Affiliations:** 1Department of Dermatology, “Iuliu Hațieganu” University of Medicine and Pharmacy, 400006 Cluj-Napoca, Romania; iuliazboras@yahoo.com (I.Z.); simonasenila@yahoo.com (S.Ș.); cosgarear@yahoo.com (R.C.); 2Department of Pathology, Regional Institute of Gastroenterology and Hepatology, 400162 Cluj-Napoca, Romania; bobe.petrushev@gmail.com (B.P.); paulacristinela@yahoo.com (P.Z.); 3Department of Pathology, “Iuliu Hațieganu” University of Medicine and Pharmacy, 400006 Cluj-Napoca, Romania; doinitacrisan@gmail.com; 4Faculty of Medicine, “Iuliu Hațieganu” University of Medicine and Pharmacy, 400012 Cluj-Napoca, Romania; zandrei75@yahoo.com; 5Department of Pharmacology, Toxicology and Clinical Pharmacology, “Iuliu Hațieganu” University of Medicine and Pharmacy, 400337 Cluj-Napoca, Romania; stefanvesa@gmail.com

**Keywords:** PRAME, immunohistochemistry, melanoma, nevus

## Abstract

PRAME (PReferentially expressed Antigen in Melanoma) immunohistochemistry has proven helpful in distinguishing malignant from benign melanocytic tumors. We studied PRAME IHC expression in 46 thin melanomas and 39 melanocytic nevi, mostly dysplastic nevi. Twenty-six percent (26.09%) of the melanomas showed diffuse PRAME staining in over 76% of the tumor cells (4+), and 34.78% of the melanomas showed PRAME expression in over 51% of the tumor cells (3+ or 4+), while 8% were entirely negative for PRAME. No melanocytic nevi were PRAME 4+ or 3+. More than half of the nevi (64%) were entirely negative for PRAME staining, and 36% of the nevi showed staining expression in 1–25% (1+) or 26–50% of the cells (2+). No nevi were stained with a color intensity of 3, while 16.67% of the melanomas were stained with this color intensity. Most nevi (78.57%) were stained with an intensity of 1. With a lower positivity threshold, sensitivity increases with still reasonable specificity. The best accuracy was obtained for the 2+ positivity threshold. In conclusion, PRAME staining helps distinguish thin melanomas from dysplastic nevi. However, the threshold of positivity should be lowered in order not to miss thin melanomas.

## 1. Introduction

PRAME (PReferentially expressed Antigen in Melanoma) immunohistochemistry (IHC) has proven its diagnostic utility in differentiating between benign melanocytic tumors and malignant melanocytic tumors [1]. Melanocytic neoplasms are classified according to the latest WHO classification of melanocytic skin tumors, 5th edition and are shown in Table 1 [2]. PRAME was mostly positive in superficial spreading melanomas (SSMs), acral melanomas (AMs), nodular melanomas (NMs) or lentigo maligna melanomas (LMMs) and only in a few desmoplastic melanomas (DMs) [3,4]. It was also observed to be positive in most melanoma in situ cases [4]. PRAME immunohistochemistry can be positive in some cases of Spitz nevi (SN) or atypical Spitz tumors (ASTs) but in a lower proportion of cases compared to spitzoid melanomas (SMs), and the proportion is lower in SMs compared to SSMs and LMMs [5,6,7].

PRAME immunohistochemistry was sometimes interpreted as positive in dysplastic nevi (DN) but in a lower proportion compared to melanomas [8]. In challenging melanocytic tumors, a higher positivity of PRAME staining was observed when compared to nevi, but a lower positivity compared to melanomas, thus supporting the histopathological result [9]. There was good concordance between PRAME IHC results in challenging melanocytic tumors and other cytogenetic test results like fluorescence in situ hybridization (FISH) and single-nucleotide polymorphism (SNP) array and between PRAME IHC results and the final diagnostic interpretation [10]. Compared to FISH testing, PRAME staining had lower sensitivity in spitzoid neoplasms and other atypical melanocytic neoplasms [11]. McAfee et al. found no statistically significant correlation between PRAME staining and FISH testing in spitzoid tumors [12].

PRAME immunohistochemistry can be used for a better margin assessment of lentigo maligna (LM) and lentigo maligna melanomas [1,13,14]. Slow Mohs micrographic surgery is the best procedure to assess the margins in LM and LMM cases [15]. Adding special immunohistochemistry like PRAME or combined immunohistochemistry PRAME/Melan A to slow Mohs micrographic surgery could help assess the margins in lentigo maligna and lentigo maligna melanoma cases [16].

Combining nuclear staining PRAME with membranous staining like Melan A and HMB-45 could be of potential use in the diagnosis of melanoma, especially in complex cases [16,17,18]. The double staining with the two melanocytic markers Melan A and HMB-45 helps to assess PRAME immunohistochemistry on melanocytes better [19]. PRAME immunohistochemistry can differentiate nodal nevi from metastatic melanomas, but it is recommended to use prior H&E and other melanocytic markers (SOX 10 or Melan A) to confirm the presence of melanocytes in the sentinel lymph node biopsy or to use double staining PRAME/Melan A [20]. PRAME staining is superior to HMB-45 in differentiating benign from malignant melanocytic tumors, but combining nuclear PRAME staining with membranous HMB-45 staining can increase specificity [21]. Combining PRAME with p16 staining has proven helpful in distinguishing between benign and malignant melanocytic lesions, with PRAME being mostly positive in malignant lesions and p16 being mostly positive in benign lesions [22].

Regarding the prognostic value of PRAME immunohistochemistry, it seems to have no impact on disease-specific survival [23]. Lo Bello et al. observed no statistically significant correlation between PRAME positivity and relapse or survival rate [24].

Our study aimed to evaluate whether PRAME immunohistochemistry can effectively differentiate thin melanomas, defined as melanomas with a Breslow index of ≤1 mm, from nevi, primarily dysplastic nevi, which are the main histopathological differential diagnoses for melanomas. There are only a few published studies that focus specifically on thin melanomas or melanomas in situ with regard to PRAME immunohistochemistry. However, some studies have included these groups alongside more advanced melanoma cases. Our research focused on a Romanian patient population, and to the best of our knowledge, no previous studies on PRAME immunohistochemistry in a Romanian population have been published.

## 2. Materials and Methods

### 2.1. Study Design and Patients

This retrospective study included 46 thin melanomas and 39 melanocytic nevi (Figure 1) diagnosed in the Department of Dermatology and Venereology of Cluj-Napoca Emergency County Hospital between 2014 and 2019. The study was approved by the Ethics Committee of Iuliu Hatieganu University of Medicine and Pharmacy Cluj-Napoca, Romania. All participants gave their informed consent. All melanocytic lesions were reviewed by a pathologist (D.C.).

### 2.2. Immunohistochemistry

The paraffin-embedded blocks were retrieved from the Department of Pathology of Cluj-Napoca Emergency County Hospital and were cut into 5 mm thick tissue sections for immunohistochemical analysis. We performed depigmentation with hydrogen peroxide 3% to remove the excessive melanin. The immunohistochemistry staining was performed with the recombinant anti-PRAME antibody [ERP20330] (ab219650) from Abcam (Cambridge, UK) on an automated Leica Bond-Max stainer platform from Leica Biosystems (Melbourne, Australia) at a 1:100 dilution using a DAB brown chromogen. We used sebaceous glands as positive internal controls, while non-melanocytic and non-sebaceous cells were used as negative internal controls.

The staining results were recorded as the percentage and intensity of immunoreactive tumor cells with nuclear labeling. No staining at all indicated 0%; staining of 1% to 25% of tumor cells was scored as 1+; staining of 26% to 50% of tumor cells was scored as 2+; staining of 51% to 75% of tumor cells was scored 3+; and staining of more than 76% of tumor cells was scored as 4+. Staining intensity was recorded as negative (0), weak (1), moderate (2) and strong (3).

### 2.3. Statistical Analysis

Statistical analysis was performed in R version 4.4.0–“Puppy Cup”. Descriptive statistics were reported for all variables. Continuous variables were presented as mean and standard, while categorical variables were reported as frequency and percentage. The Student’s *t*-test was used for the comparison of continuous variables between two groups, while the chi-square test was used to test for differences in frequency between groups. Sensitivity, specificity, negative predictive value (NPV), positive predictive value (PPV) and accuracy parameters were computed for different thresholds of PRAME staining to assess the discriminatory capacity between nevi and melanomas. A *p*-value of less than 0.05 was considered statistically significant.

## 3. Results

We examined 46 melanomas for PRAME immunohistochemistry in this study: 16 stage 0 melanomas, 16 stage IA melanomas and 14 stage IB melanomas. The group of melanomas included 38 superficial spreading melanomas, two acral lentiginous melanomas (ALMs), three lentigo maligna and three lentigo maligna melanomas. Eighteen patients were female, and twenty-eight patients were male. Patient age ranged between 26 and 85 years, with a mean age of 57.56 and a median age of 60. The Breslow thickness ranged between 0 and 1 mm, with a mean Breslow thickness of 0.45 mm and a median Breslow thickness of 0.5 mm. Regarding localization, 24 melanomas were located on the trunk, eight on the head and neck regions, eight on the lower limbs and six on the upper limbs. Of all melanomas, 31 were in a horizontal growth phase, while 15 were in a vertical growth phase. The mitotic rate ranged from 0 to 12 mitosis/mm^2^, with a mean of 1.41 mitosis/mm^2^ and a median of 0 mitosis/mm^2^.

Twenty-six (26.09%) of the melanomas showed diffuse PRAME staining in over 76% of the tumor cells (4+) (Figure 2, Figure 3 and Figure 4), and 8.7% showed PRAME staining in 51 to 75% of the cells (3+), while 8% were entirely negative for PRAME. In total, 34.78% of the melanomas showed PRAME staining in over 51% of the tumor cells. Considering only melanoma in situ cases, from 16 melanoma in situ cases, only 2 stained in over 76% (4+) of the tumor cells, meaning only 12.5% of the cases and 1 case stained 3+, accounting for 6.25% of the cases. However, all melanomas showed slight staining, with the majority in the 2+ group (11/16), accounting for 68.75% of the cases.

In comparison to melanomas, we examined 39 melanocytic nevi: 36 dysplastic nevi, one dermal nevus, one halo nevus and one acral nevus. One nevus was a high-grade dysplastic nevus; 29 nevi were low-grade dysplastic nevi, and six nevi had unspecified grades of dysplasia. From the dysplastic nevi, two were in acral location. No nevi showed positive PRAME staining in over 76% (4+) or 51 to 75% of the cells (3+). A total of 64% of the nevi were entirely negative for PRAME staining, and 36% of the nevi stained in 1–25% of the cells (1+) or 26–50% (Table 2). There was no statistically significant difference in color intensity of PRAME staining between the nevi and the melanoma group. No nevi stained with a color intensity of 3, while 16.67% of the melanomas stained with a color intensity of 3. Most nevi (78.57%) stained with an intensity of 1. The high-grade dysplastic nevus stained 1+ with a color intensity of 1. The dermal and the halo nevus did not show PRAME staining in any cells (0). The acral nevus stained in 1–25% (1+) of the cells.

In the melanoma patient group, melanomas with vertical growth were positive for PRAME immunohistochemistry in over 76% of the tumor cells (4+), more frequently than other groups in terms of staining percentage (*p* < 0.05) (Table 3). Different variables like sex, age, localization, histopathological subtype, stage, ulceration, regression, Breslow index, mitotic rate, personal or family history of melanoma or development on pre-existent nevus showed no statistically significant association with PRAME staining.

In the LM group, all three cases stained 26–50% (2+) of the tumor cells, while in the LMM group, one case stained 51–75% (3+), one case stained 1–25% (1+) of the tumor cells, and one case was completely negative.

In our study, sensitivity for a cutoff value of >76% (4+) or >51% (3+ and 4+) was low, 26% and 35%, respectively, but specificity was 100% for both (Table 4). The cutoff value of >26% (2+) provided the best accuracy and Youden’s index.

Considering a lower positivity threshold and interpreting 0 and 1+ as PRAME-negative and 2+, 3+ and 4+ as PRAME-positive, 76.09% of the melanomas were PRAME-positive, while 84.61% of the nevi were negative.

Regarding survival, 45 patients out of 46 were alive by July 2024, while one patient died of another cause.

## 4. Discussion

### 4.1. PRAME Staining in Melanoma

In the study conducted by Lezcano et al., 83.2% of the melanomas were diffusely PRAME-positive (4+) [1], while in the Gassenmeier et al. study conducted on thin melanomas (Breslow index ≤ 1 mm), the melanomas were PRAME-positive (4+) in 58.6% of them [25]. Our study included only thin melanomas with a Breslow index ≤ 1 mm, similarly to the study conducted by Gassenmeier et al., and our percentage of PRAME-positive melanoma cases was 26.09%, almost half compared to the previously mentioned study [25]. The mean Breslow thickness in their study was 0.7 mm (range 0.3–1.0) [25], higher than our mean Breslow thickness of only 0.45 mm (range 0–1.0), which is probably the reason for our lower proportion of positive cases, or probably because Gassenmeier et al. used a different dilution (1:50) of the antibody compared to our dilution (1:100), and they used another clone (clone QR005, DCS, Hamburg, Germany). In addition, they included stage III and stage IV metastasizing and non-metastasizing melanomas, while our study included only non-metastasizing stage 0 (in situ) and stage I melanomas. In our study, 16/46 melanomas were in situ. In Lezcano et al.’s study [1], the melanoma in situ cases showed positive PRAME staining (4+) in 93.8% of the cases compared to our study group of melanoma in situ cases in which only 12.5% of the cases stained 4+. However, although they used the same antibody as ours, they did not mention the dilution. Moreover, the entirely negative melanoma cases were similar to those of other studies, 8.7%, compared to the Lezcano et al. [1] study in which 8% of the melanoma cases were entirely negative for PRAME IHC. The difference is that we observed many cases in the intermediate staining groups (3+ and 2+).

Parra et al. observed no impact on disease-specific survival regarding PRAME immunohistochemistry, but they observed a positive correlation between PRAME-positive staining in melanomas and a higher mitotic rate (*p*  =  0.047) [23]. We observed a positive correlation between PRAME positivity and vertical growth in melanomas (*p* = 0.027) but no statistically significant correlation with the mitotic rate. Out of 46 melanoma patients, 45 are alive and one patient died of another cause.

In other studies, lentigo maligna and lentigo maligna melanomas have similar PRAME expressions compared to superficial spreading melanomas [1]. Tio et al. observed a higher expression of PRAME in lentigo maligna melanomas than in lentigo maligna [26]. In our study, no lentigo maligna or lentigo maligna melanoma case stained in over 76% of the tumor cells (4+), but all three lentigo maligna cases and 2/3 lentigo maligna melanoma cases stained for PRAME, but in a lower proportion of the cells. However, the included cases were too few.

In acral melanocytic tumors, PRAME staining proved to help distinguish benign from malignant lesions. Still, the proportion of PRAME-positive cases was higher for invasive melanomas than for melanomas in situ [27]. It proved its diagnostic utility in both subungual and non-subungual acral melanomas [28]. PRAME staining proved superior in distinguishing acral melanomas from acral nevi compared to p16 staining [29]. In our study, both acral lentiginous melanoma cases were stained in 26–50% (2+) of the tumor cells with a 2 and 1 intensity score, respectively. If considering a 4+ or 3+ positivity threshold, both cases would be negative, suggesting that PRAME staining is not always helpful in distinguishing benign from malignant lesions. If a positivity threshold of 2+ was considered, both cases would be positive.

### 4.2. PRAME Staining in Melanocytic Nevi

No nevi in our study were positive for PRAME staining in over 50% of the cells. Although 36% of melanocytic nevi were focally positive 1+ or 2+ for PRAME staining, a higher percentage compared to the study conducted by Lezcano et al., where only 13.6% were described [1], this can also be due to the fact that our study mainly included DN. In the study conducted by Cazzato et al., 96.4% of the nevi were PRAME-negative or had a score of 1+ [30]. The results in our study are similar, with 84.61% PRAME-negative nevi or with a score of 1+. DN can sometimes be diffusely positive for PRAME immunohistochemistry in over 76% of the tumor cells (4+). Turner et al. found a 10% positivity of DN in over 75% of the cells (4+) [8], but in our study, no nevi showed diffuse positive staining. Rasic et al. observed a higher diffuse positivity of PRAME staining in high-grade dysplastic nevi compared to low-grade dysplastic nevi and common nevi [21]. In the study carried out by Innocenti et al., PRAME staining could differentiate between high-grade dysplastic nevi and cutaneous melanomas or between low-grade dysplastic nevi and cutaneous melanomas, but not between high-grade dysplastic nevi and low-grade dysplastic nevi [31]. Lezcano et al. found a single melanocytic nevus to be diffusely positive for PRAME (4+), which was described as a Spitz nevus, and more nevi showed focal positive PRAME staining (1+ or 2+) [1]. Raghavan et al. also found one Spitz nevus to be diffusely positive (4+) for PRAME staining, but no dysplastic nevi, recurrent nevi, mitotically active nevi or traumatized nevi showed diffuse positive PRAME staining (4+) [7].

### 4.3. Interpretation of PRAME Staining

Although the first studies considered PRAME staining positive if it was diffusely present in over 76% of the tumor cells (4+) [1], more recently, Kunc et al. suggested in their meta-analysis that PRAME positivity should be interpreted as both 4+ (>75% of the cells) and 3+ (51–75% of the cells) cases in clinical practice due to better sensitivity with reasonable specificity [32]. In the study conducted by Rawson et al., 35% of the melanomas showed 4+ staining, an outcome closer to our study. If 3+ and 4+ represented positive staining, the percentage of PRAME-positive melanomas was 64%. No nevi showed 4+ staining, similarly to our study, while only 4% showed 3+ staining, with both cases being dysplastic nevi. Thus, the author suggested considering PRAME as positive for 3+ staining (present in 50–76% of the tumor cells) or 4+ staining (present in over 76% of the tumor cells [3]. As in our study no nevi showed 3+ or 4+ staining, we can say that the previous statement is also suitable for our research, although we found more melanomas with 4+ staining than 3+. Raghavan et al. considered PRAME as positive if staining was present in over 60% of the tumor cells in order to improve sensitivity. Their study included atypical melanocytic proliferations of indeterminate behavior and atypical Spitz tumors. In both groups, the expression of PRAME was low; only one atypical melanocytic proliferation was positive for PRAME staining in 10% of the cells, and one atypical Spitz tumor was positive for PRAME staining in over 60% of the cells [7]. O’Connor et al. suggested that the results should be interpreted as favoring nevus if PRAME staining is present in <25% of the cells, noncontributory if PRAME staining is present in 26–75% of the cells and favoring melanoma if PRAME staining is present in >76% of the cells. In their study, most melanomas were in situ and pT1a, like in our research. They found 64% of the melanomas positive for PRAME staining in >76% of the cells compared to 26% of the melanomas in our study. However, staining was performed on another automated platform (BenchMark ULTRA IHC/ISH System, Roche Diagnostics, Indianapolis, IN, USA) using PRAME EPR20330 antibody from Biocare Medical [33]. Alomari et al. described positive staining cases with ‘hotspot’ staining, defined as cases with diffuse staining (over 75% of the tumor cells) in at least two adjacent high-power fields [9]. Warbasse et al. considered cases 2+, 3+ and 4+ positive for PRAME staining with low sensitivity (29.6%) on a series of spitzoid and challenging melanocytic neoplasms [11]. Umano et al. recorded the percentage of positive cells, staining intensity (1+: slight positivity, 2+: moderate positivity and 3+: intense positivity) and the location of positive cells (junctional or intradermal) [34]. Forchhamer et al. observed a lower proportion of PRAME-positive melanoma cases in the pediatric population than in the adult population, suggesting that age might be considered when interpreting PRAME staining in melanomas [35]. In our study, no nevi stained 3+ or 4+, while the majority of melanomas stained 2+, 3+ or 4+ with the best accuracy and Youden’s index for the 2+, 3+ and 4+ groups; therefore, we suggest considering at least 3+ as a threshold for positivity, but further studies are needed with a higher number of cases included. The results should be interpreted according to the dilution of the antibody, the technique used, the stainer vendor and the previous results of the given histopathological laboratory so as not to miss any thin melanomas.

## 5. Conclusions

PRAME immunohistochemistry is a powerful diagnostic tool for distinguishing melanocytic nevi from thin melanomas, but the interpretation should be performed carefully. Combining more immunohistochemistry antibodies would probably give more specific results with better sensitivity and specificity, but further studies are needed. We suggest a lower positivity threshold for PRAME staining to avoid missing any thin melanomas, but differences could appear between different histopathological laboratories.

## Figures and Tables

**Figure 1 diagnostics-14-02015-f001:**
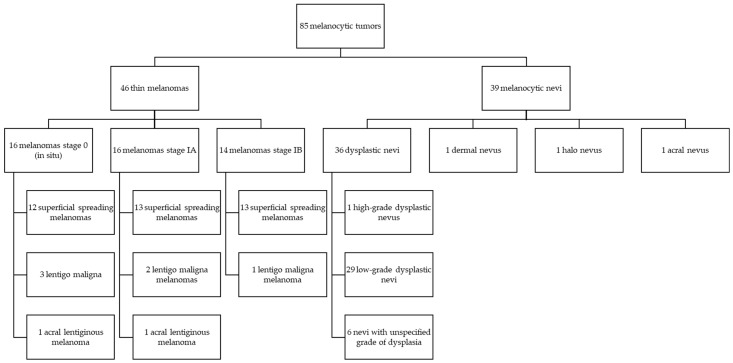
Study design (Melanocytic tumor distribution).

**Figure 2 diagnostics-14-02015-f002:**
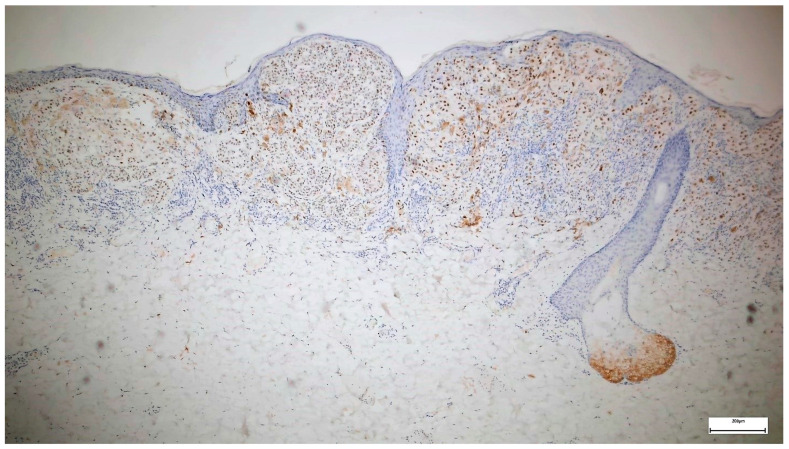
PRAME immunohistochemistry in melanoma (staining in over 76% of the cells 4+, staining intensity 3)—sebaceous gland as positive internal control 50×.

**Figure 3 diagnostics-14-02015-f003:**
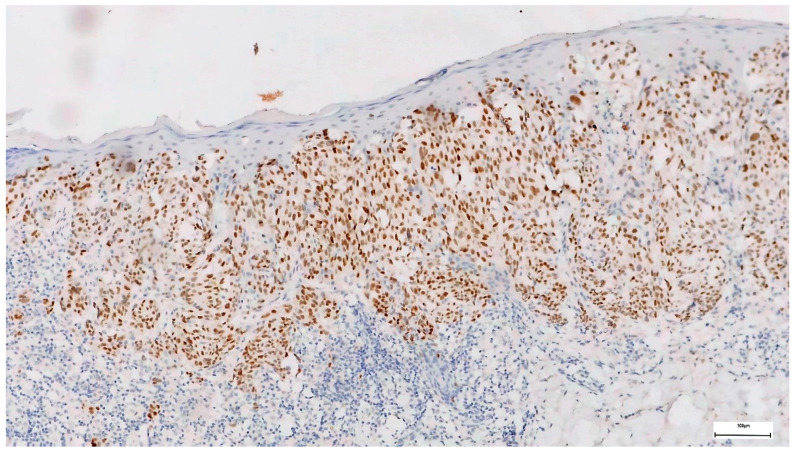
PRAME immunohistochemistry in melanoma (staining in over 76% of the cells 4+, staining intensity 3) 100×.

**Figure 4 diagnostics-14-02015-f004:**
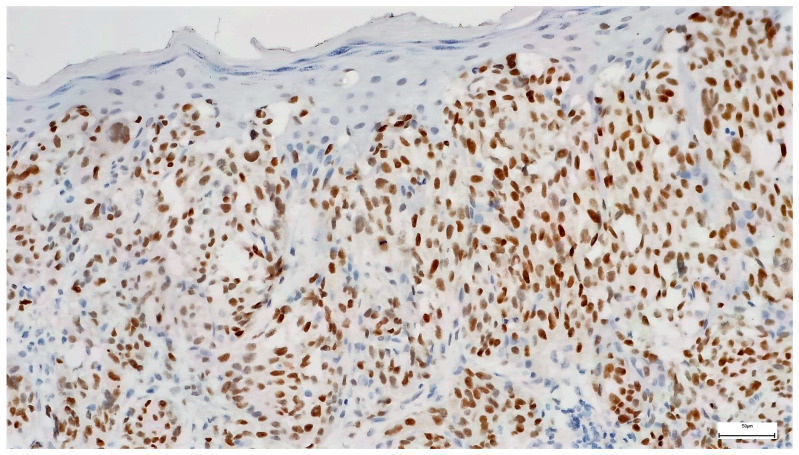
PRAME immunohistochemistry in melanoma (staining in over 76% of the cells 4+, staining intensity 3) 200×.

**Table 1 diagnostics-14-02015-t001:** Classification of melanocytic neoplasms according to WHO classification of melanocytic skin tumors, 5th edition.

Melanocytic Neoplasms	Subtypes	
Melanocytic Neoplasms in Intermittently Sun-Exposed Skin	Nevi	Junctional, compound and dermal nevi
		Simple lentigo and lentiginous melanocytic nevus
		Dysplastic nevus
		Nevus spilus
		Special-site nevus (of the breast, axilla, scalp and ear)
		Halo nevus
		Meyerson nevus
		Recurrent nevus
		Combined nevus
	Melanocytomas	WNT-activated deep-penetrating/plexiform melanocytoma (nevus)
		Pigmented epithelioid melanocytoma
		BAP1-inactivated melanocytoma
		MITF pathway-activated melanocytic tumor
	Melanomas in intermittently sun-exposed skin	Melanoma on skin with low cumulative sun damage (low CSD); includes superficial spreading melanoma
Melanocytic Neoplasms in Chronically Sun-Exposed Skin	Lentigo maligna melanomas	
	Desmoplastic melanomas	
Spitz Tumors	Spitz nevi	Pigmented spindle cell nevus (Reed nevus)
		Spitz nevus
	Spitz melanocytomas	Spitz melanocytoma (atypical Spitz tumor)
	Spitz melanomas	
Melanocytic Tumors in Acral Skin	Acral nevi	
	Acral melanomas	
Genital and Mucosal Melanocytic Tumors	Mucosal and genital nevi	Melanosis
		Genital nevus
	Mucosal melanomas	
Blue Nevi and Related Tumors	Blue nevi and melanocytoses	Nevus of Ito and nevus of Ota
		Congenital dermal melanocytosis
		Blue nevus
	Melanomas arising in blue nevi	
Congenital Melanocytic Tumors	Congenital nevi	Congenital melanocytic nevus
		Proliferative nodules in congenital melanocytic nevus
	Melanomas arising in congenital nevi	Melanoma arising in giant congenital nevus
Ocular and Central Nervous System (CNS) Melanocytic Tumors	Conjunctival melanocytic tumors	Conjunctival nevus
		Conjunctival melanocytic intraepithelial lesion
		Conjunctival melanoma
	Uveal melanocytic tumors	Uveal melanocytoma
		Uveal melanoma
	CNS melanocytic tumors	Diffuse meningeal melanocytic neoplasms: melanocytosis and melanomatosis
		Circumscribed meningeal melanocytic neoplasms: melanocytoma and melanoma
Nodular, Nevoid and MetaStatic Melanomas	Nodular and other melanomas	Nodular melanoma
		Nevoid melanoma
		Dermal melanoma
	Metastatic melanomas	Melanoma metastatic to the skin
		Melanoma metastatic to other organs

**Table 2 diagnostics-14-02015-t002:** Comparison between PRAME immunohistochemistry in melanomas vs. nevi.

		Nevi	Melanomas	*p*-Value
Staining percentage N (%)	0% (0)	25 (64.1)	4 (8.7)	<0.001
	1–25% (1+)	8 (20.51)	7 (15.22)	
	26–50% (2+)	6 (15.38)	19 (41.3)	
	51–75% (3+)	0 (0)	4 (8.7)	
	>76% (4+)	0 (0)	12 (26.09)	
Staining percentage N (%)	0, 1+, 2+	39 (100)	30 (65.22)	<0.001
	3+, 4+	0 (0)	16 (34.78)	
Staining percentage N (%)	0, 1+, 2+,3+	39 (100)	34 (73.91)	0.001
	4+	0 (0)	12 (26.09)	
Color intensity N (%)	1	11 (78.57)	24 (57.14)	0.202
	2	3 (21.43)	11 (26.19)	
	3	0 (0)	7 (16.67)	

**Table 3 diagnostics-14-02015-t003:** Comparison between PRAME staining in melanomas in over 76% of the tumor cells (4+) and under 75% of the tumor cells (0, 1+, 2+ and 3+) according to clinical and histopathological variables.

Variable			Melanoma PRAME Staining Groups 0, 1+, 2+ and 3+ N = 34	Melanoma PRAME Staining Group 4+ N = 12	*p*-Value
Sex N (%)		F	14 (41.18)	4 (33.33)	0.632 ^$^
		M	20 (58.82)	8 (66.67)	
Age [years] Median ± sd			58.68 ± 13.82	54.42 ± 15	0.374 ^#^
Localization N (%)		Head and neck	8 (23.53)	0	0.111 ^$^
		Trunk	16 (47.06)	8 (66.67)	
		Upper limb	3 (8.82)	3 (25)	
		Lower limb	7 (20.59)	1 (8.33)	
Histopathological subtype N (%)		SSM	26 (76.47)	12 (100)	0.332 ^$^
		LMM	3 (8.82)	0	
		LM	3 (8.82)	0	
		ALM	2 (5.88)	0	
Vertical growth N (%)		No	26 (76.47)	5 (41.67)	**0.027** ^$^
		Yes	8 (23.53)	7 (58.33)	
Stage N (%)		In situ	14 (41.18)	2 (16.67)	0.167 ^$^
		IA	12 (35.29)	4 (33.33)	
		IB	8 (23.53)	6 (50)	
Ulceration N (%)		No	34 (100)	11 (91.67)	0.089 ^$^
		Yes	0	1 (8.33)	
Regression N (%)		No	27 (79.41)	10 (83.33)	0.768 ^$^
		Yes	7 (20.59)	2 (16.67)	
Breslow index Median ± sd			0.4 ± 0.36	0.61 ± 0.35	0.080 ^#^
Mitotic rate Median ± sd			1.38 ± 2.63	1.5 ± 1.57	0.885 ^#^
Family history of melanoma	No		32 (94.12)	12 (100)	
	Yes		2 (5.88)	0 (0)	0.390
Personal history of melanoma	No		28 (82.35)	12 (100)	
	Yes		6 (17.65)	0 (0)	0.119
Nevus-associated melanoma	No		26 (76.47)	7 (58.33)	
	Yes		8 (23.53)	5 (41.67)	0.230

^$^ Chi-Square test; ^#^ *t*-test.

**Table 4 diagnostics-14-02015-t004:** Diagnostic values by different cutoff values for PRAME.

Cutoff	Sensitivity (%)	Specificity (%)	Positive Predictive Value (%)	Negative Predictive Value (%)	Accuracy (%)	Youden’s Index
>76% (4+)	26%	100%	100%	53%	60%	0.26
>51% (3+, 4+)	35%	100%	100%	56%	64%	0.35
>26% (2+, 3+, 4+)	76%	85%	85%	75%	80%	0.61
>1% (1+, 2+, 3+, 4+)	91%	64%	75%	86%	79%	0.55

## Data Availability

The datasets used and/or analyzed during the current study are available from the corresponding author on reasonable request.

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
