# Peer review of "PRAME Immunohistochemistry in Thin Melanomas Compared to Melanocytic Nevi"

_diagnostics, 2024, doi:10.3390/diagnostics14182015_

Round 1

Reviewer 1 Report

Comments and Suggestions for Authors

Iulia ZboraÈ™ et al present a very interesting and well-performed study on IHC for PRAME. In my opinion, this study increases and amplifies our knowledge on this crucial diagnostic tool (especially in relation to the cut-offs of PRAME to adopt for the best performance), for which I recommend the acceptance after the following “minor revisions”:

-As this is not a standard nomenclature, please consider “early melanoma”/”early” melanoma/”early-stage” melanoma (with quotation marks);

-“Combining PRAME with p16 staining has proved helpful in distinguishing between benign and malignant melanocytic lesions, with PRAME being mostly positive in malignant lesions and p16 being mostly positive in benign lesions”.
This is true, but adding melanocyte markers (double stains with PRAME) also helps to assess PRAME on melanocytes alone, leaving out keratinocytes, lymphocytes, fibroblasts and other inflammatory cells (PMID: 31633488; PMID: 35682589);

-Please, specifiy if the two acral melanoma were acral lentiginous melanoma;

-“From the dysplastic nevi, 3 were acral nevi. One nevus was a high-grade dysplastic nevus; 30 nevi were low-grade dysplastic nevi, and eight nevi had unspecified dysplasia
According to WHO 2023 Classification of Skin Tumors, “acral” and “dysplastic” nevus are two different categories. Please, specify if the 3 acral nevi were acral or dyspalstic ones (if they are dysplastic naevi in acral location, they are not “really acral” from a classification side). Please, specify what do you mean with eight nevi had unspecified dysplasia;

-I found very interesting (and needing investigation) that the adopted cut-off of PRAME should be discussed. Besides, I find almost impossible to adopt a “universal” cutt-off of PRAME for any subtype of melanocytic lesions (Spitz, blue, SSM, acral, meningeal, etc.). Please, cite and discuss the other papers that had highlighted this issue (PMID: 32700786; PMID: 36912431; PMID: 36942814);

Reviewer 2 Report

Comments and Suggestions for Authors

The manuscript represents an assessment of PRAME expression in early melanomas and melanocytic nevi using immunohistochemistry in a Romanian cohort of patients. The search for new diagnostic markers for melanoma is important; however, this work raises several questions.

First, the data presentation is poorly structured, which makes the article difficult to read, especially for readers unfamiliar with the classification of melanoma or melanocytic nevi. Incorporating abbreviations or study design visualization could improve clarity.

Second, the novelty of this work is unclear. The aim of the study was "to test whether PRAME immunohistochemistry can differentiate early melanomas, defined as stage 0 and stage I, from nevi, primarily dysplastic nevi, in Romanian patients." Nevertheless, it remains uncertain why this cohort of patients may differ significantly from the numerous studies of PRAME expression that have already been published involving other populations. The authors should explicitly emphasize the novelty of their study.

Finally, the authors do not explain some of the results obtained, such as:

- In comparison to the study by Gassenmeier et al., there is a lower percentage of PRAME-positive melanoma cases.

- In comparison to the study by Lezcano et al., there is a lower percentage of PRAME-positive cases (4+) of melanoma in situ.

- In comparison to the study by Parra et al., there is no correlation with the mitotic rate, etc.

The statement "we suggest 2+ as a threshold for positivity for higher sensitivity" requires additional evidence. It appears that the number of samples used in this study does not support such conclusions.

Round 2

Reviewer 2 Report

Comments and Suggestions for Authors

The authors provided a complete and exhaustive answer to all questions. In my opinion, the added information has improved the manuscript considerably.

However, authors should include the abbreviations used in the «Introduction» in the «Discussion» section, which will also help improve the readability of the manuscript (lines 261-266).
